# New Emerging Therapies Targeting PI3K/AKT/mTOR/PTEN Pathway in Hormonal Receptor-Positive and HER2-Negative Breast Cancer—Current State and Molecular Pathology Perspective

**DOI:** 10.3390/cancers17010016

**Published:** 2024-12-24

**Authors:** Liu Liu, Stephanie L. Graff, Yihong Wang

**Affiliations:** 1Department of Pathology and Laboratory Medicine, Rhode Island Hospital and Brown University Health, Providence, RI 02903, USA; lliu@brownhealth.org; 2Legorreta Cancer Center, Warren Alpert School of Medicine, Brown University, Providence, RI 02903, USA; sgraff1@brownhealth.org; 3Division of Medical Oncology, Rhode Island Hospital and Brown University Health, Providence, RI 02903, USA

**Keywords:** breast cancer, genomic alteration, PI3K/AKT/mTOR/PTEN pathway, molecular pathology

## Abstract

Hormone receptor-positive, HER2-negative breast cancer is the most prevalent subtype of breast cancer, and also represents the majority of metastatic breast cancer cases. Dysregulation of the PI3K/AKT/mTOR pathway in breast cancer plays a critical role in resistance to standard therapies. In recent years, several inhibitors targeting the PI3K/AKT/mTOR pathway have been approved by the FDA to treat ER-positive, HER2-negative, high-risk and metastatic breast cancer, such as alpelisib (a PI3K inhibitor) and everolimus (an mTOR inhibitor), often combined with standard endocrine therapy like fulvestrant or aromatase inhibitors, particularly when a PIK3CA mutation is present in the tumor; AKT inhibitors like capivasertib may also be considered in some cases depending on the specific genetic profile of the tumor. However, clinicians face growing challenges in understanding the mechanism behind the genome events associated with this pathway and selecting the most effective therapy. This review focuses on the current state of the novel therapeutic options targeting the PI3K/AKT/mTOR/PTEN pathway and discusses the molecular mechanism and genomic testing interpretation from a molecular pathology perspective.

## 1. Introduction

Breast cancer (BC) is a heterogeneous group of diseases. Based on molecular markers, it can be divided into four subtypes: hormonal receptor (HR) positive Luminal A, HR-positive Luminal B, human epidermal growth factor receptor 2 (HER2) positive, and triple negative. Approximately 65–70% of the BCs are HR-positive and HER2-negative [1,2,3,4]. Moreover, HR-positive and HER2-negative BCs account for 65–70% of all metastatic BC cases [3].

In recent years, improved technology, and the discovery of molecular markers, as well as the identification of mutations in key cellular pathways has led to the development of a growing number of novel and targeted therapy options. Among them are cyclin-dependent kinase 4 and 6 (CDK4/6) inhibitors, oral selective estrogen receptor down regulators, antibody–drug conjugates (ADCs), immune checkpoint inhibitors, and phosphatidylinositol 3-kinase (PI3K)/AKT/mTOR pathway inhibitors. Those novel agents have been used in clinical trials and clinical practice for treating HR+, HER2- high-risk BCs in adjuvant, neoadjuvant, and metastatic settings, and have resulted in significant improvements in patient’s quality of life and survival. With the increasing number of treatment options, clinicians face challenges in selecting the best treatment for the patient, appropriate drug sequencing, and understanding drug interactions.

Understanding the genomic alterations of each BC case is of crucial importance. Molecular pathology and cell biology pathways, genomic profile, circulating tumor DNA, cancer stem cells, and biomarker research have been some of the key areas of development that are transforming the field of BC treatment. Improving the outcomes for HR+/HER2– BC requires collaborative efforts involving oncologists and molecular pathologists. This review focuses on the current state of the novel therapeutic options of the PI3K/AKT/mTOR/PTEN pathway and discusses the molecular mechanism and genomic testing interpretation from a molecular pathology perspective. This review aims to help clinicians and pathologists understand the characteristics of various genomic alterations and when they may be detected to help treatment selection and optimization for BC patients.

## 2. PI3K/AKT/mTOR Pathway

The PI3K/AKT/mTOR pathway is the most studied pathway in BC. Over 70% of BCs carry an alteration in this pathway leading to the activation of this signaling pathway [5]. Abnormal activation of the PI3K/AKT/mTOR pathway drives excessive cell proliferation and resistance to apoptosis and contributes to the initiation and progression of tumors [6]. Dysregulation of the PI3K/AKT/mTOR pathway has an impact on resistance to endocrine therapy, PARP inhibitors, chemo- and targeted therapies, leading to cancer progression and poorer outcomes, and blockage of this pathway can enhance drug sensitivity and reverse resistance [5,7,8,9]. PI3K pathway inhibitors targeting key components of the pathway are listed in Table 1.

The PI3K pathway is activated by receptor tyrosine kinase (RTK) stimulation. The key players and regulators in this pathway involve PI3K, AKT, mTOR, TSC1/2, and PTEN (Figure 1). A dynamic complex network is formed in cancer cells by regulating many substrates and crosstalk between components within the pathway or communicating with other signaling pathways.

Dysregulation of the PI3K/AKT pathway in BC is commonly caused by RTK activation, such as in the case of HER2 amplification, KRAS mutations, and PI3K alterations such as PIK3CA-activating mutations, and less often, PIK3CB, PIK3R1, PIK3R2 mutations, AKT1-activating mutations, inactivation of tumor suppressor genes such as PTEN, and TSC1/2 [10]. Dysregulation of the PI3K/AKT/mTOR pathway by activating mutations in PIK3CA and AKT1 and/or inactivating PTEN are most frequently observed in HR+/HER2- BCs, which are found to be mutually exclusive (GENIE breast cancer cohort, Appendix A). The luminal-type BCs had the highest frequency of PIK3CA mutations, approximately 45% in luminal A, and 29% in luminal B. Frequencies of PTEN mutations/losses in luminal A and B are about 13% and 24%, respectively. However, PTEN loss is found to be more common using the immunohistochemistry method due to epigenetic modifications. AKT1-activating mutations are less frequent, up to 7.4% across all BCs studied [10].

### 2.1. PI3Ks

PI3Ks are present in nearly all cell membranes regulate signaling, membrane trafficking, and metabolic pathways. They can be divided into three PI3K classes (classes 1, 2, and 3) based on additional protein domain presentations and their interactions with regulatory subunits [11]. The class 1 PI3Ks are heterodimers composed of a catalytic and a regulatory subunit. The catalytic subunit can be one of the four subunits (p110α, β, δ, or γ) encoded by genes PIK3CA, PIK3CB, PIK3CD, and PIK3CG, respectively. The regulatory subunit can be one of the p85s (p85α, p85β, and p85γ), encoded by PIK3R1, PIK3R2, and PIK3R3, respectively. The class 1 subtype has been well studied and is tightly associated with tumor progression in humans [12]. Among these, catalytic subunits p110α and p110β and regulatory subunits p85α and p85β are broadly expressed in BCs [13].

PIK3CA mutations are the most common genetic alterations in the PI3K/AKT/mTOR pathway and can be identified across BC subtypes. The luminal-type BCs had the highest frequency of PIK3CA mutations, especially the luminal A subtype. A comprehensive molecular study on 800 BC patients revealed that PIK3CA is one of the most frequently mutated genes. About 45% of Luminal A and 29% of Liminal B BCs have PIK3CA alterations [14]. Mutations in PIK3CA predominantly occur in two hot-spot regions: the helical domain (E542, E545, Q546) and kinase domain (H1047). These mutations act as gain-of-functions of PI3K activity, resulting in aberrant activation of AKT/mTOR downstream signaling for tumorigenesis [15,16]. PIK3CA gain/amplification occurs as a less frequent event and is present in 8.7% of BCs [17]. Up to 25% of advanced BCs with PIK3CA mutations have a second mutation on the same allele (i.e., in *cis*). Tumors with double PIK3CA mutations have a more active PI3K enzyme and might be more responsive to PI3K inhibitors than those with single mutations [18].

PI3K inhibitors have been developed to inhibit one or more of the PI3K components and can be categorized into three groups: pan-PI3K inhibitors, isoform-specific inhibitors, and dual PI3K-mTOR inhibitors (Table 1).

Pan-PI3K inhibitors are designed to target all PI3K catalytic subunits (PIK3CA, PIK3CB, PIK3CD and PIK3CG). This type of inhibitor has a wide range of molecular targets and, therefore, is predicted to be effective on a broader spectrum may be a better choice for dysregulated PI3K tumors with heterogeneous molecular alterations. However, the toxicities can be high. Clinical trials BELLE-3 of pan-PI3K inhibitor buparlisib with ESR1 antagonist fulvestrant in patients with HR+/HER2− advanced or metastatic BCs showed statistically significantly longer progression-free survival (PFS) in the treatment group vs. placebo; however, due to high rates of grade 3–4 adverse events, further development of buparlisib in BC was halted [19,20].

Isoform-specific PI3K inhibitors include alpelisib and inavolisib. The majority of the drugs in this category selectively target the class 1 PI3K catalytic subunit α (p110α) encoded by PIK3CA. Alpelisib has been approved by the FDA for PIK3CA-mutated HR+ HER2- BC [21], along with its companion diagnostic test, therascreen^®^ PIK3CA RGQ PCR Kit (QIAGEN GmbH, QIAGEN Strasse 1, Hilden, Germany). More recently, the FDA has approved inavolisib for PIK3CA mutated HR+ HER2- BC along with its companion diagnostic device, the FoundationOne Liquid CDx assay (Foundation Medicine, Inc. Cambridge, MA USA) [22]. Both alpelisib and inavolisib target PIK3CA mutations. In contrast to alpelisib, which had a stable affinity profile across all PIK3CA hotspot mutations, inavolisib affinity was slightly higher in the presence of H1047R mutation but not the E542K or E545K [23]. Inavolisib was approved based on the results of the INAVO120 trial, testing the addition of the PIK3CA targeting agent in the first-line metastatic setting, which is likely to alter the way PIK3CA testing is considered in clinical practice, as compared to previous PIK3CA agents in later lines of therapy [24]. A clinical trial evaluating inavolisib plus fulvestrant versus alpelisib plus fulvestrant in patients with HR+, HER2–, PIK3CA-mutated locally advanced or metastatic BC is ongoing (NCT05646862, INAVO121).

Both PI3K and mTOR belong to the PI3K-related kinases (PIKK) superfamily. Due to the structural similarity between the PI3K and catalytic pocket of mTOR, dual PI3K/mTOR inhibitors have been developed targeting both mTOR complexes (mTORC1 and mTORC2) andall four PI3K catalytic subunits at the same time, inhibiting the PI3K pathway both upstream and downstream of AKT. This group of drugs has shown better anti-cancer effects than single targeting inhibitors [25]. Dual PI3K-mTOR inhibitor gedatolisib has been granted a fast-track designation by the FDA as a potential therapeutic option in patients with HR+/HER2- metastatic BC who experienced disease progression on CDK4/6 therapy [26]. A Phase 3 study (VIKTORIA-1) evaluating gedatolisib plus fulvestrant with and without palbociclib in patients with advanced BC is ongoing [27].

Mutations in other genes in the PI3K, while less frequent, can serve as important biomarkers and explain the resistance mechanism. Activating mutations in catalytic subunit PIK3CB alters the interaction between p110 and p85 subunits and confers resistance to the pan-PI3K inhibitor; such resistance could be overcome by applying downstream inhibitors targeting AKT or mTOR complexes by way of experiments in the PTEN-depletion BC cell line [28]. In some cases, mutation testing can provide biomarkers for alternate targeted therapies that would otherwise be missed. For example, recurrent truncated mutations in regulatory subunit PIK3R1, one of the p85 subunits, are gain-of-function events abolishing the binding to the p110α subunit, increasing JNK and ERK phosphorylation, and activating the MAPK pathway. These PIK3R1 mutations are insensitive to PI3K/ATK inhibitors but confer sensitivity to MEK and JNK inhibitors [29].

### 2.2. AKT and PTEN

AKT serine/threonine kinase is another key component of the PI3K pathway. It has three isoforms, encoded by AKT1, AKT2, and AKT3, respectively, and share high sequence homolog and conservative protein function domains. AKT1 and AKT2 are ubiquitously expressed, while AKT3 is mainly present in neural cells [30]. In cancer cells, AKT1 is involved in proliferation and growth, promoting tumor initiation and suppressing apoptosis, whereas AKT2 regulates cytoskeleton dynamics, favoring invasion and metastasis [30]. Overexpression of AKT1 and AKT2 are frequent events in BC that enhance tumor cell survival [31]. AKT3 is less studied due to its limited expression in the breast. However, aberrant activation of AKT3 has been associated with tumor progression in BC [32].

All three isoforms have a common hotspot mutation, E17K, in the protein kinase domain. This mutation promotes the localization of AKTs to the plasma membrane through a PI3K-independent mechanism and activates the signal pathway [33]. AKT1 E17K mutation is restricted to HR+ BC and is significantly enriched in metastatic BC compared to primary BC [34]. AKT2/3 E17K mutations are rare. Gene amplification or, in rare cases, AKT fusions can be observed in the cancer cells as the activating event. AKT2 and AKT3 amplifications are observed more frequently (0.85%, 0.70%), and AKT1 amplification was less (0.48%) among AKT isoforms in the GENIE breast cancer cohort, as seen in Appendix A).

Phosphatase and tensin homolog (PTEN) is a tumor suppressor gene that encodes for dual-specificity lipid and protein phosphatase, which downregulates the PI3K/AKT signaling. Dephosphorylation of PIP3 by PTEN negatively regulates the PI3K/AKT pathway, suppressing the growth and survival signals [35]. Loss of PTEN function in BC leads to PI3K signaling activation.

Loss of PTEN can be caused by a variety of changes at different levels, including single nucleotide changes (SNVs) such as missense and nonsense mutations, small insertion/deletions (indels), copy number deletions, transcriptional down-regulation by miRNA regulation, silencing of the gene through promoter methylation, and post-translational alterations affecting the PTEN protein [36]. Loss of heterozygosity at the PTEN locus was reported in nearly 40–50% of BCs, whereas the loss of PTEN function due to PTEN mutations was detected in 5–10% of BCs, with frameshift representing the most frequent mechanism [37]. Loss of the PTEN gene is associated with poor prognosis in HR+ HER2- BC and represents a highly aggressive, treatment-refractory group of diseases [38]. Loss of PTEN strongly activates AKT through activation of p110β, leading to resistance to drugs that target p110α, such as alpelisib [39].

### 2.3. mTOR

Mammalian target of rapamycin (mTOR) is a serine and threonine kinase downstream of the PI3K/AKT signaling pathway. It forms two complexes, mTOR complex 1 (mTORC1) and mTOR complex 2 (mTORC2). mTORC1 comprises mTOR, raptor, GβL, and deptor, while mTORC2 is composed of mTOR, Rictor, GβL, PRR5, deptor, and SIN1 [40]. mTORC1 mainly regulates cell growth and metabolism, while mTORC2 mainly controls cell proliferation and survival. In BC, mTORC1 also promotes BC proliferation by inducing new lipid synthesis [40].

Mutations in mTOR itself are rare in BC and mutation status in mTOR is not required for using mTOR inhibitors [41]. As mTOR is more downstream in the pathway, targeting mTOR could overcome CDK4/6 inhibitor resistance in HR+ BC [42]. Everolimus is the mTOR inhibitor approved by the FDA in HR+ HER2- BC, regardless of mTOR mutation status. It binds to the intracellular protein FKBP-12 encoded by FKBP1A and forms a complex inhibiting the mTORC1 activity [43]. Dual PI3K/mTOR inhibitors are under development to target both PI3Ks and mTORs, as mentioned in the PI3K section in this paper.

## 3. Other PI3K Pathway-Related Genes

In addition to the PI3K components for which targeted therapy has been approved, other genes in the PI3K pathway play critical roles and may be potential therapy targets, such as the TSC1/2 complex, PDK1, and RET.

### 3.1. TSC1/2

TSC Complex Subunit 1 (TSC1) and TSC Complex Subunit 2 (TSC2) are tumor suppressor genes. TSC1 and TSC2 form a complex TSC1/2, act as GTPase-activating proteins, and negatively regulate mTORC1 signaling. Hypermethylation in promoter regions of TSC1 is observed in BCs and is associated with an unfavorable clinical outcome [44]. TSC2 is the hub of multiple signaling pathway networks. Mutations in TSC2 were significantly enriched in metastatic BC compared to primary BC, suggesting that it is associated with tumor progression [34].

### 3.2. PDK1

The phosphoinositide-dependent protein kinase-1 (PDK1) encodes a serine/threonine protein kinase and is another key component of the PI3K pathway. Upon pathway activation, PIP3 binds to AKT1 and PDK1 to bring them close and facilitate PDK1 phosphorylating and the activation of AKT1 [12]. PDK1 overexpression and amplifications are common events reported in 20% of the BCs [45].

### 3.3. RET

Ret proto-oncogene (RET) receptor tyrosine kinase activation can induce the PI3K/AKT pathway. RET is a direct target for ER. ER upregulated RET and was associated with tamoxifen resistance in HR+ BC [46]. RET protein overexpression has been observed in 40–60% of BCs without gene amplification, whereas RET genomic alterations are less frequent in only 1.2% of cases [47]. In one study, among 121 reported alterations, 67% were RET gene amplifications, 20% were point mutations, and 13% were gene rearrangements [47]. While RET rearrangements and amplifications are significantly associated with ER-negative BCs, RET missense point mutations were associated with ER-positive BCs. RET inhibitors have not been a focus of clinical trials in BCs. However, much preclinical data have been collected that supports their essential role as a future therapy target in BCs [48].

## 4. Pathway Crosstalk

The PI3K pathway discussed above is a simplified model; many PI3K pathway components interact with other pathway networks. The PI3K pathway combines many other pathways and forms a complex network in the cancer cell signaling. The synergic effect of these pathways should be considered during target therapy development and selection.

One well-studied example is the interplay between the PI3K and ER pathways in the BCs (Figure 2). ER-positive BC is considered as a type of estrogen-dependent cancer; standard treatments typically involve hormone therapies that either block estrogen production or prevent estrogen from interacting with ER. However, the PI3K pathway can activate ER in the absence of estrogen by phosphorylating ER through AKT1 [49]. As a result, genome alterations such as PIK3CA mutation and the loss of PTEN expression, which upregulates the PI3K pathway, can also promote estrogen-independent ER activation and render breast cancer cells resistant to endocrine therapy. ER promotes the transcription of genes that enhance the PI3K pathway, including RTKs, ligands, and adaptors [50]. ER can also bind directly to p85α to increase PI3K signaling [51]. When the ER pathway is inhibited in BC patients receiving endocrine therapy, the PI3K pathway is enhanced, leading to therapy resistance [52]. On the other hand, inhibition of the PI3K pathway increases ER-mediated cell signaling for survival [53]. Based on these findings, PI3K pathway inhibitors show greater efficacy when combined with fulvestrant, an estrogen receptor antagonist. Combination therapy provides the best effect for cancer treatment if toxicity is manageable. The FDA has approved alpelisib, in combination with fulvestrant for PIK3CA mutated HR+ HER2- BC, and capivasertib in combination with fulvestrant for HR+ HER2- BC with alterations in PIK3CA/AKT1/PTEN.

Crosstalk between the PI3K and cell cycle pathways also influences targeted therapy efficacy in BC. Upregulating the cell cycle pathway with hyperphosphorylation Rb inhibits mTORC2-mediated AKT signaling, further stimulating cell growth [54,55]. A study shows that CDK4/6 inhibitor-resistant cell lines remain sensitive to mTORC1/2 inhibition [56]. Therefore, targeting mTOR could be an option for HR+ CDK4/6 inhibitor treatment-resistant BC [42].

## 5. Building an HR+/HER2- Breast Cancer Specific Molecular Profile

Immunohistochemistry (IHC) is a routine test in pathology laboratories, using antibodies targeting certain antigens in formalin-fixed paraffin-embedded (FFPE) tissue for cell classification, tumor diagnosis and prognosis. Gene expression profiling assays such as Prosigna^®^ PAM50 (Veracyte, Inc., South San Francisco, CA, USA) have been widely used for recurrent risk assessment in BC. In the molecular era, molecular testing is offered to metastatic BC patients experiencing progression after initial, standard treatments. Second-line treatment options in advanced or metastatic HR+ HER2- BC are highly dependent on the actionable genomic biomarker information in the tumor or circulating tumor DNA (ctDNA) of the patient.

For example, therascreen^®^ PIK3CA RGQ PCR Kit (QIAGEN GmbH, QIAGEN Strasse 1, Hilden, Germany) is a companion diagnostic test approved by the FDA for alpelisib in treating PIK3CA-mutated HR+ advanced BC [57]. The therascreen targets 11 mutations in the PIK3CA gene, with a limit of detection (LoD) ranging from 2.4 to 14.0% for individual targeted PIK3CA mutations in tissue specimens. Although not completely identical, clinical laboratories that perform NGS often cover variable mutation profiles overlapping with the FDA-approved test, as oncoReveal™ Solid Tumor Panel (Pillar Biosciences, Inc. Natick, MA USA) can detect all 11 PIK3CA mutation variants, with additional variants from 48 genes with LoD targeted to as low as 1% using tissue specimen. The FoundationOne Liquid CDx assay (Foundation Medicine, Inc. Cambridge, MA USA) was recently approved as a companion diagnostic device for inavolisib in PIK3CA mutated HR+ HER2- BC [22]. The FoundationOne Liquid CDx assay is a blood-based test to report substitutions and indels in 311 genes, and the most comprehensive FDA-approved liquid biopsy on the market.

Given the nature of tumor heterogeneity, molecular testing with new advanced technologies, applied in clinics, including comprehensive next-generation sequencing (NGS), allows for obtaining broad genetic information on cancer-related regions at once. Liquid biopsies utilizing ctDNA have also become popular as a less invasive method to monitor tumor recurrence or metastasis, enabling detection of disease progression at a much earlier stage than conventional clinical relapse [58,59].

For HR+ HER2- BC seeking a second-line option, a targeted NGS BC panel is very useful to tell whether the tumor harbors an actionable mutation in critical BC-related genes, including ESR1, PIK3CA, AKT1, PTEN, as well as additional genes in the PI3K pathway or other synergetic pathways. If hereditary BC is suspected and BRCA1/2 germline testing is considered, a germline specimen, such as blood, buccal swab, or skin biopsy, needs to be considered. In rare situations, RNA sequencing for fusion detection may be considered to rule out rare RET or NTRK fusions, which have potential alternative targeted therapies. If microsatellite instability (MSI) and tumor mutation burden (TMB) need to be assessed, an NGS panel including MSI and TMB could be selected. Homologous recombination deficiency (HRD) status for potential PARP inhibitors is also a valuable marker that can be covered by an NGS panel or separated molecular assay [60].

NGS assay is a popular test that allows the simultaneous detection of alterations from thousands of genome regions at a variant allele frequency as low as 1–5%. The NGS assays can also be designed and optimized to assess the MSI, TMB, and HRD status and capture copy number variations (CNV). Epigenetic profiling plays a role in regulating gene expression and necessitates a separate assay for capturing, such as a methylation array. As a short-read sequencing platform with a size limit of a couple of hundred base pairs, NGS may not reliably call large-size indels and is not an appropriate assay for structural variants. Long-read sequencing retains the methylation information in the genome, and technology with the ability to sequence fragments with sizes as large as tens of thousands of base pairs will be promising for complex genome alterations in future clinical diagnosis [61,62].

Most solid tumor testing using FFPE tissue faces the challenges of dealing with damaged DNA/RNA molecules, resulting in poor quality data which is difficult to interpret. Limiting tumor tissue on needle biopsies, especially small gauge needle biopsies for metastatic BC, is another challenge faced by pathologists on a daily basis. Although a comprehensive large NGS panel that covers more biomarkers may be preferred, a small biopsy that is not sufficient for a large NGS assay could fail to return any biomarker information. In this case, a small, actionable, amplicon-based NGS assay may have a higher chance of success.

Therefore, it is critical for clinicians to understand that the careful selection of assays, based on clinical information and specimen conditions, is of utmost importance. This process is not one that should be undertaken in isolation. Instead, a collaborative effort is strongly recommended between the oncologists and pathologists. With appropriate testing, using a combination of conventional methods and new advanced technologies such as comprehensive genomic profiling in treatment-resistant BC patients, we can identify potential biomarkers explaining the resistance mechanism. This not only provides opportunities for customized targeted therapy with better responses but also paves the way for a future of cancer treatment that is tailored to meet individual patient’s needs, offering new hope in the fight against cancer.

## 6. Conclusions

The PI3K pathway is upregulated in HR+ BCs. Hyperactivation of the PI3K pathway plays an important role in endocrine resistance in BC. Different types of genome alterations from different genes involved in the PI3K pathway can lead to signaling dysregulation. Several agents targeting components in the PI3K pathway have been approved by the FDA or are showing promising results in clinical trials. Recognizing tumor characteristics and identifying tumor subgroups which potentially respond to a specific targeted agent from heterogenous BC is a big step forward in personalizing breast cancer care.

Decision-making is even more challenging when clinicians must compare which tests to order and how to correctly interpret molecular reports. In this era, molecular testing and reporting are variable and less standardized. From a molecular pathology perspective, we need to establish how molecular testing are implemented to ensure biomarker testing is carried out correctly and effectively. We need to stay on top of technological advancement, help the clinician choose the adequate tissue and blood sample for the correct molecular testing, and interpret the molecular biomarkers and their roles in certain pathways and tumor network signaling. Now is the time for oncologists and molecular pathologists to seize the amazing opportunities to work together for the best management of breast cancer patients.

## Figures and Tables

**Figure 1 cancers-17-00016-f001:**
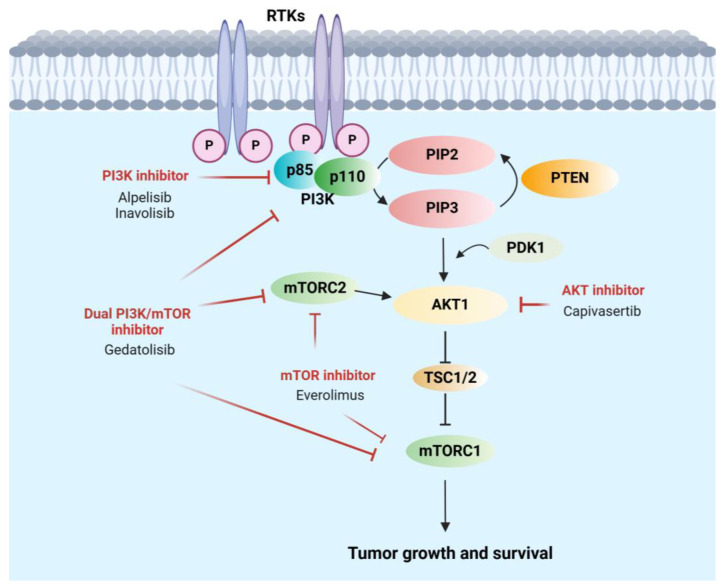
The PI3K pathway and FDA-approved, or fast-track designation-granted PI3K inhibitors. The PI3K pathway is activated by receptor tyrosine kinase (TRK) stimulation, which recruits PI3K to the cell membrane and induces PIP2 phosphorylation to PIP3, promoting AKT activation by PDK1. Activation of mTORC is led by AKT-induced inhibition of the TSC1/2 complex. PTEN serves as a negative regulator by functioning as a PIP3 phosphatase and converting PIP3 to PIP2. mTOR complexes regulate both upstream and downstream of AKT. mTORC2 promotes AKT activation from upstream of the pathway; AKT inhibits TSC1/2 complexes, which inhibit its downstream target mTORC1 complex. (Figure created in BioRender.com).

**Figure 2 cancers-17-00016-f002:**
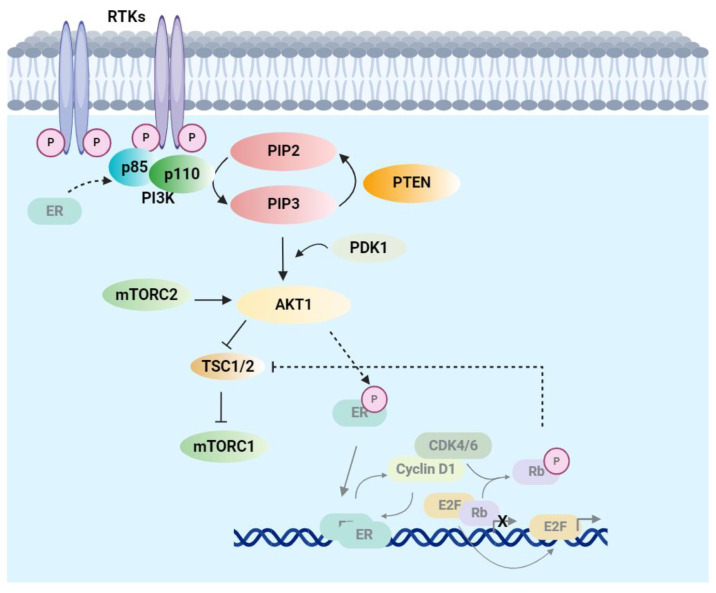
PI3K, cell cycle, and ER pathways crosstalk in HR+ HER- BC. AKT1 phosphorates ER and activates the ligand-independent ER pathway. ER can bind p85 and increase PI3K pathway signaling. The CCND1/CDK4/6 complex phosphorylates Rb and can negatively regulate the TSC1/2 complex to activate mTORC1. A dynamic complex network is formed in cancer cells due to the regulation of many substrates and crosstalk between components within the pathway or the communication with other signaling pathways. ER and cell cycle pathway components are present in gray. The interplay between PI3K and cell cycle or ER pathway is indicated by dotted lines. (Figure created in BioRender.com).

**Table 1 cancers-17-00016-t001:** Selected PI3K pathway inhibitors for advanced or metastatic HR+, HER2- breast cancer.

Drugs (Brand Name)	FDA Status	Granted Year	Target	Mutation	Clinical Trials	Trial Registration Number	Patient Population
Inavolisib (Itovebi)	Approved	2024	PIK3CA	PIK3CA mutations	INAVO120, INAVIO121	NCT04191499 NCT05646862	ER+, HER2- mBC or laBC, +ET,
Capivasertib (Truqap)	Approved	2023	pan-AKT	mutations in PIK3CA/AKT1/PTEN	CAPItello-292	NCT04862663	ER+, HER2- mBC or laBC, +ET, +/-CDK4/6i
Gedatolisib	Fast track designation	2022	dual PI3K and mTOR	not required	VIKTORIA-1	NCT05501886	ER+, HER2- mBC +CDK4/6i, +AI
Alpelisib (Piqray)	Approved	2019	PIK3CA	PIK3CA mutations	SOLAR-1	NCT02437318	ER+, HER2- mBC +ET, +/-CDK4/6i
Everolimus (Afinitor)	Approved	2012	mTOR	not required	BOLERO-2	NCT00863655	ER+, HER2- mBC +AI
Buparlisib	No	NO	pan-class 1 PI3K	not required	BELLE-2, BELLE-3	NCT01610284 NCT01633060	ER+, HER2- mBC or laBC

Abbreviation: laBC, local advanced breast cancer; mBC, metastatic breast cancer; AI, aromatase inhibitors; ET, endocrine therapy.

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
