# Peer review of "New Emerging Therapies Targeting PI3K/AKT/mTOR/PTEN Pathway in Hormonal Receptor-Positive and HER2-Negative Breast Cancer—Current State and Molecular Pathology Perspective"

_cancers, 2024, doi:10.3390/cancers17010016_

Round 1
Reviewer 1 Report
Comments and Suggestions for Authors
In this review authors discussed the emerging therapies Targeting PI3K/AKT/mTOR/PTEN pathway in Hormonal Receptor Positive and HER2 Negative breast Cancer. Overall, the manuscript is well written, and it provides valuable information in the field of breast cancer research. I have few comments which should be addressed:
1. Line 11-13: This review focuses on the current state of the novel therapeutic options of the PI3K/AKT/mTOR/PTEN pathway and discusses the molecular mechanism and genomic testing interpretation from a molecular pathology perspective.
In the above sentence authors should write therapeutic options for targeting PI3K/AKT/mTOR/PTEN pathway.
2. Line 357-360: Although many new biomarkers are emerging recent year due to advanced technology allowing studies at genome-wide scale, due to the complexity within the signal pathway and interplay between different pathways, developing new effective treatment remains challenge.
The above sentence should be rephrased.
3. Line 351-352: Identifying tumors potentially response to these agents from heterogenous BC are very important.
This sentence should be rephrased.
4. There are few minor English language corrections in the manuscript. Authors should pay attention to this and make all the necessary corrections.
Author Response
REVIEWER1
Comments 1: Line 11-13: This review focuses on the current state of the novel therapeutic options of the PI3K/AKT/mTOR/PTEN pathway and discusses the molecular mechanism and genomic testing interpretation from a molecular pathology perspective.
In the above sentence authors should write therapeutic options for targeting PI3K/AKT/mTOR/PTEN pathway.
Theraeutic options for targeting PI3K/AKT/mTOR/PTEN pathway has been added in the modified Simple Summary Section.
Response 1: Thank you for pointing out, we agree with this comment. We have updated the simple summary with the therapeutic options. Full revised summary is included here and updated in the manuscript:
Hormone receptor (HR)-positive, HER2-negative breast cancer is the most prevalent subtype of breast cancer and also represents the majority of metastatic breast cancer cases. Dysregulation of the PI3K/AKT/mTOR pathway in breast cancer plays a critical role in resistance to standard therapies. In recent years, several inhibitors targeting PI3K/AKT/mTOR pathway have been approved by FDA to treat ER-positive, HER2-negative, high-risk and metastatic breast cancer, such as alpelisib (a PI3K inhibitor) and everolimus (an mTOR inhibitor), often combined with standard endocrine therapy like fulvestrant or aromatase inhibitors, particularly when a PIK3CA mutation is present in the tumor; AKT inhibitors like capivasertib may also be considered in some cases depending on the specific genetic profile of the tumor. However, clinicians face growing challenges in understanding the mechanism behind the genome events associated with this pathway and selecting the most effective therapy. This review focuses on the current state of the novel therapeutic options targeting the PI3K/AKT/mTOR/PTEN pathway and discusses the molecular mechanism and genomic testing interpretation from a molecular pathology perspective.
Comments 2: Line 357-360: Although many new biomarkers are emerging recent year due to advanced technology allowing studies at genome-wide scale, due to the complexity within the signal pathway and interplay between different pathways, developing new effective treatment remains challenge.
The above sentence should be rephrased.
Response 2: Thank you for pointing out, we agree with this comment. This has been rephrased in the manuscript. The revised ‘section 6. Conclusion’ is included here and updated in the manuscript:
The PI3K pathway is upregulated in HR+ BCs. Hyperactivation of the PI3K pathway plays an important role in endocrine resistance in BC. Different types of genome alterations from different genes involved in the PI3K pathway can lead to signaling dysregulation. Several agents targeting components in the PI3K pathway have been approved by the FDA or are showing promising results in clinical trials. Recognizing tumor characteristics and identifying tumor subgroups potentially responding to a specific targeted agent from heterogenous BC is a big step forward in personalizing breast cancer care.
Decision-making is even more challenging when clinicians must compare which tests to order and how to correctly interpret molecular reports. In this era, molecular testing and reporting are variably and less standardized. From a molecular pathology perspective, we need to establish how molecular testing gets implemented to ensure biomarker testing is done correctly and effectively. We need to stay on top of technological advancement, help the clinician choose the adequate tissue and blood sample for the correct molecular testing, and interpret the molecular biomarkers and their role in certain pathways and the tumor network signaling. Now is the time for oncologists and molecular pathologists to seize the amazing opportunities to work together for the best management of breast cancer patients.
Suggestion:
Although many new biomarkers have emerged recently due to advanced technology allowing studies at genome-wide scale, because of the complexity within the signal pathway and interplay between different pathways, developing new effective treatment remains challenge.
Comments 3: Line 351-352: Identifying tumors potentially response to these agents from heterogenous BC are very important.
This sentence should be rephrased.
Response 3: Thank you, this has been rephrased in the manuscript. The revised ‘section 6. Conclusion’ is included here and updated in the manuscript:
The PI3K pathway is upregulated in HR+ BCs. Hyperactivation of the PI3K pathway plays an important role in endocrine resistance in BC. Different types of genome alterations from different genes involved in the PI3K pathway can lead to signaling dysregulation. Several agents targeting components in the PI3K pathway have been approved by the FDA or are showing promising results in clinical trials. Recognizing tumor characteristics and identifying tumor subgroups potentially responding to a specific targeted agent from heterogenous BC is a big step forward in personalizing breast cancer care.
Decision-making is even more challenging when clinicians must compare which tests to order and how to correctly interpret molecular reports. In this era, molecular testing and reporting are variably and less standardized. From a molecular pathology perspective, we need to establish how molecular testing gets implemented to ensure biomarker testing is done correctly and effectively. We need to stay on top of technological advancement, help the clinician choose the adequate tissue and blood sample for the correct molecular testing, and interpret the molecular biomarkers and their role in certain pathways and the tumor network signaling. Now is the time for oncologists and molecular pathologists to seize the amazing opportunities to work together for the best management of breast cancer patients.
Reviewer 2 Report
Comments and Suggestions for Authors
After reading it, I found the article very interesting, it raises important aspects of personalized therapy for hormonal receptor positive and HER2 negative breast cancer. The authors focused on changes in the PI3K/AKT/mTOR/PTEN pathway. They are trying to prove that understanding genomic changes in breast cancer is very important for clinicians who make decisions about therapy. Analysis of these changes also allows for the discovery of new targets for new therapies. The manuscript also presents selected PI3K signaling pathway inhibitors selected based on new clinical trials for advanced or metastatic HR+, HER2- breast cancer (Table 1 and Figure 1).
The paper also presents FDA-approved molecular diagnostic tests for the detection of PIK3CA mutations (therascreen PIK3CA RGQ PCR Kit, diagnostic device FoundationOne Liquide CDx assay) and selected drug programs still in the final phase of clinical trials.
In addition to PI3K, they highlighted the importance of molecular changes in the serine-threonine kinase AKT located in the PI3K signaling pathway as well as the importance of single nucleotide changes in the gene encoding the PTEN protein and mTORC2 as well some other PI3K patheay related genes such as the TSC1/2 complex, PDK1, and receptor tyrosine kinase ret proto-oncogene (RET).
It also highlighted the increasing need to develop molecular profiles in breast cancer, which have already entered clinical practice, such as PAM50 (Prosigna), therascreen® PIK3CA RGQ PCR Kit (QIAGEN Manchester, Ltd.) as well as the latest solutions using sequencing technology (NGS).
The presented data are supported by very well-selected literature, which will allow the reader to expand on the topics raised by the authors.
In my opinion, the work is a very good study, summarizing and highlighting the latest trends in personalized therapy in Hormonal Receptor Positive and HER2 Negative Breast Cancer.
After reading it, I can only have minor comments related to the editing of the manuscript e.g. – page 1
line 29 is ….. and HER2-negative. [1, 2]. – should be ….and HER2-negative [1, 2].
Line 30 is …cases[3]. – should be …cases [3].
Line 176 is ….BC[31]. – should be ….BC [31].
In conclusion, I would like to recommend the manuscript submitted for review for publication in Cancers.
Author Response
Comments 4. There are few minor English language corrections in the manuscript. Authors should pay attention to this and make all the necessary corrections.
After reading it, I found the article very interesting, it raises important aspects of personalized therapy for hormonal receptor positive and HER2 negative breast cancer. The authors focused on changes in the PI3K/AKT/mTOR/PTEN pathway. They are trying to prove that understanding genomic changes in breast cancer is very important for clinicians who make decisions about therapy. Analysis of these changes also allows for the discovery of new targets for new therapies. The manuscript also presents selected PI3K signaling pathway inhibitors selected based on new clinical trials for advanced or metastatic HR+, HER2- breast cancer (Table 1 and Figure 1).
The paper also presents FDA-approved molecular diagnostic tests for the detection of PIK3CA mutations (therascreen PIK3CA RGQ PCR Kit, diagnostic device FoundationOne Liquide CDx assay) and selected drug programs still in the final phase of clinical trials.
In addition to PI3K, they highlighted the importance of molecular changes in the serine-threonine kinase AKT located in the PI3K signaling pathway as well as the importance of single nucleotide changes in the gene encoding the PTEN protein and mTORC2 as well some other PI3K patheay related genes such as the TSC1/2 complex, PDK1, and receptor tyrosine kinase ret proto-oncogene (RET).
It also highlighted the increasing need to develop molecular profiles in breast cancer, which have already entered clinical practice, such as PAM50 (Prosigna), therascreen® PIK3CA RGQ PCR Kit (QIAGEN Manchester, Ltd.) as well as the latest solutions using sequencing technology (NGS).
The presented data are supported by very well-selected literature, which will allow the reader to expand on the topics raised by the authors.
In my opinion, the work is a very good study, summarizing and highlighting the latest trends in personalized therapy in Hormonal Receptor Positive and HER2 Negative Breast Cancer.
After reading it, I can only have minor comments related to the editing of the manuscript e.g. – page 1
line 29 is ….. and HER2-negative. [1, 2]. – should be ….and HER2-negative [1, 2].
Line 30 is …cases[3]. – should be …cases [3].
Line 176 is ….BC[31]. – should be ….BC [31].
In conclusion, I would like to recommend the manuscript submitted for review for publication in Cancers.
Response: Thank you for pointing out, we agree with this comment. These changes in lines 29, 30 and 176 have been updated in the manuscript.
Reviewer 3 Report
Comments and Suggestions for Authors
In this review article, Liu and colleagues focus on providing a comprehensive overview of the current therapeutic options for targeting PI3K/AKT/mTOR/PTEN pathway in HR+/HER2- breast cancer patients. The authors provide a detailed description of the molecular pathway and the current therapeutic options available to interfere with the pathway at different levels depending on the context. In general, I believe that this review can serve as a valuable resource for the scientific community. However, I would encourage the authors to enhance its content by providing a more comprehensive discussion of their perspective. This will enable it to distinguish itself from the numerous other reviews that address similar topics. Here are some specific comments:
Line 29: can the authors identify newer references for this?
Line 126: write the full form of PFS
Lines 260-274:
the explanation of the crosstalk between ER and the PI3K pathway is a little confusing and feels counterintuitive the way it is explained. If the PI3K pathway activates ER and the ER activates the PI3KA pathway, why is the inhibition of one of the two components increasing the other one and creating resistance? Please explain better this feedback mechanism and rephrase this part.
Can the authors cite additional and more recent references to explain this crosstalk?
Summary section:
The summary section at the end needs a profound revision. The text contains numerous grammatical errors, and the sentence structure is often challenging to comprehend. In addition, instead of a summary section, it would be important for the authors to provide a discussion that provides their unique point of view of what are the currently most effective therapeutic strategies, the challenges involving resistance mechanisms and patient stratification, and the future directions that the scientific community should aim to improve.
Comments on the Quality of English LanguageThe main text is well written, but I found the summary section difficult to read.
Author Response
REVIEWER3
In this review article, Liu and colleagues focus on providing a comprehensive overview of the current therapeutic options for targeting PI3K/AKT/mTOR/PTEN pathway in HR+/HER2- breast cancer patients. The authors provide a detailed description of the molecular pathway and the current therapeutic options available to interfere with the pathway at different levels depending on the context. In general, I believe that this review can serve as a valuable resource for the scientific community. However, I would encourage the authors to enhance its content by providing a more comprehensive discussion of their perspective. This will enable it to distinguish itself from the numerous other reviews that address similar topics. Here are some specific comments:
Comments: Line 29: can the authors identify newer references for this?
Response: Thank you for pointing out, we agree with this comment. This has been updated in the manuscript with two more recent references: PMID: 39179659 and PMID: 36939293.
Comments: Line 126: write the full form of PFS
Response: Thank you for pointing out, we agree with this comment. Full form of PFS has been updated in the manuscript.
Comments: Lines 260-274:
the explanation of the crosstalk between ER and the PI3K pathway is a little confusing and feels counterintuitive the way it is explained. If the PI3K pathway activates ER and the ER activates the PI3KA pathway, why is the inhibition of one of the two components increasing the other one and creating resistance? Please explain better this feedback mechanism and rephrase this part.
Can the authors cite additional and more recent references to explain this crosstalk?
Response: Thank you for pointing out, we agree with this comment. The revised paragraph is included here and updated in the manuscript:
One well-studied example is the interplay between PI3K and ER pathways in the BCs. ER-positive BC is considered a type of estrogen-dependent cancer; standard treatments typically involve hormone therapies that either block estrogen production or prevent estrogen from interacting with ER. However, the PI3K pathway can activate ER in the absence of estrogen by phosphorylating ER through AKT1 [49]. As a result, genome alterations such as PIK3CA mutation and loss of PTEN expression which upregulate the PI3K pathway, can also promote estrogen-independent ER activation and render breast cancer cells resistant to endocrine therapy. ER promotes the transcription of genes that enhance the PI3K pathway, including RTKs, ligands, and adaptors [50]. ER can also bind directly to p85α to increase PI3K signaling [51]. When the ER pathway is inhibited in BC patients receiving endocrine therapy, the PI3K pathway is enhanced, leading to therapy resistance [52].
On the other hand, inhibition of the PI3K pathway increased ER-mediated cell signaling for survival [53]. Based on these findings, PI3K pathway inhibitors show greater efficacy when combined with fulvestrant, an estrogen receptor antagonist. Combination therapy provides the best effect for cancer treatment if toxicity is manageable. The FDA approved alpelisib in combination with fulvestrant for PIK3CA mutated HR+ HER2- BC and capivasertib in combination with fulvestrant for HR+ HER2- BC with alterations in PIK3CA/AKT1/PTEN.
Comments: Summary section:
The summary section at the end needs a profound revision. The text contains numerous grammatical errors, and the sentence structure is often challenging to comprehend. In addition, instead of a summary section, it would be important for the authors to provide a discussion that provides their unique point of view of what are the currently most effective therapeutic strategies, the challenges involving resistance mechanisms and patient stratification, and the future directions that the scientific community should aim to improve.
Response: Thank you for pointing out, we agree with this comment. The revised ‘section 6. Conclusion’ is included here and updated in the manuscript:
The PI3K pathway is upregulated in HR+ BCs. Hyperactivation of the PI3K pathway plays an important role in endocrine resistance in BC. Different types of genome alterations from different genes involved in the PI3K pathway can lead to signaling dysregulation. Several agents targeting components in the PI3K pathway have been approved by the FDA or are showing promising results in clinical trials. Recognizing tumor characteristics and identifying tumor subgroups potentially responding to a specific targeted agent from heterogenous BC is a big step forward in personalizing breast cancer care.
Decision-making is even more challenging when clinicians must compare which tests to order and how to correctly interpret molecular reports. In this era, molecular testing and reporting are variably and less standardized. From a molecular pathology perspective, we need to establish how molecular testing gets implemented to ensure biomarker testing is done correctly and effectively. We need to stay on top of technological advancement, help the clinician choose the adequate tissue and blood sample for the correct molecular testing, and interpret the molecular biomarkers and their role in certain pathways and the tumor network signaling. Now is the time for oncologists and molecular pathologists to seize the amazing opportunities to work together for the best management of breast cancer patients.
Round 2
Reviewer 3 Report
Comments and Suggestions for Authors
The authors have addressed and resolved all the comments. I have no further comments to make.